# Exploring the Prognostic Role of Neurofilaments and SEMA3A in Multiple Sclerosis Progression

**DOI:** 10.3390/ijms26178750

**Published:** 2025-09-08

**Authors:** Zbyšek Pavelek, Ondřej Souček, Jan Krejsek, Ilona Součková, Andrea Popovičová, David Matyáš, Lukáš Sobíšek, Michal Novotný

**Affiliations:** 1Department of Neurology, Faculty of Medicine in Hradec Králové, Charles University, 50005 Hradec Králové, Czech Republic; andrea.popovicova@fnhk.cz (A.P.); david.matyas@fnhk.cz (D.M.);; 2Department of Neurology, University Hospital Hradec Králové, 50005 Hradec Králové, Czech Republic; 3Department of Clinical Immunology and Allergology, University Hospital Hradec Králové, 50005 Hradec Králové, Czech Republic; ondrej.soucek@fnhk.cz (O.S.); jan.krejsek@fnhk.cz (J.K.); ilona.souckova@fnhk.cz (I.S.); 4Novartis, s.r.o., 14000 Prague, Czech Republic; lukas.sobisek@yahoo.com

**Keywords:** multiple sclerosis, immunity, functional capacity, neuroimmunology

## Abstract

The transition from relapsing–remitting multiple sclerosis (RRMS) to secondary progressive multiple sclerosis (SPMS) is characterized by an increasing neurodegenerative component. Identifying biomarkers that distinguish these disease stages is crucial for early diagnosis and treatment optimization. This study aimed to compare serum levels of progranulin, interleukin-6 (IL-6), semaphorin 3A (SEMA3A), and neurofilaments between RRMS and SPMS patients and to investigate their correlation with clinical characteristics, including disability measured by the Expanded Disability Status Scale (EDSS). This observational study included 118 MS patients (63 RRMS and 55 SPMS). Serum biomarker levels were measured using an enzyme-linked immunosorbent assay (ELISA). Statistical analyses included group comparisons using non-parametric tests and correlation analyses using Pearson’s correlation coefficient with multiple testing corrections. While demographic and clinical parameters significantly differed between groups (*p* < 0.001), biomarker levels showed no statistically significant differences (*p* > 0.05). However, in SPMS patients, SEMA3A correlated positively with neurofilaments (r = 0.359, *p* = 0.007), and progranulin correlated with IL-6 (r = 0.354, *p* = 0.008). No significant biomarker correlations with EDSS were found. Although absolute biomarker levels did not distinguish RRMS from SPMS, specific biomarker correlations may reflect processes relevant to disease progression and warrant further longitudinal validation.

## 1. Introduction

Multiple sclerosis (MS) is a complex neuroinflammatory disorder, characterized primarily by distinct clinical phases, namely relapsing–remitting (RRMS) and secondary progressive (SPMS) forms [1]. A critical area of ongoing MS research involves identifying biomarkers capable of differentiating between these disease stages and predicting clinical progression [2]. Several previous studies have examined the relevance of various biomarkers in reflecting underlying neurodegenerative and inflammatory processes in MS [3,4,5].

Neurofilaments, particularly neurofilament light chains (NfLs), have emerged as robust markers of axonal injury and neurodegeneration across neurological diseases, including MS [6]. Elevated serum neurofilament levels have consistently been associated with increased disease severity and progression, especially in SPMS patients [6]. Similarly, semaphorin 3A (SEMA3A), known for its role in axonal guidance and the inhibition of axonal regeneration, has been investigated as a potential biomarker associated with neurodegeneration and disease progression in MS [7].

Interleukin-6 (IL-6), a pro-inflammatory cytokine, has also been widely studied for its role in MS, with elevated IL-6 levels generally correlating with increased disease activity and clinical deterioration [8,9]. Another molecule of interest, progranulin, has been recognized for its immunomodulatory effects, playing roles in both inflammatory and neurodegenerative pathways. Recent evidence suggests progranulin could be involved in modulating the chronic inflammatory environment and influencing neuronal survival in neurodegenerative conditions, including progressive stages of MS [10,11]. However, existing research has shown inconsistent results concerning the utility of these biomarkers in clearly distinguishing RRMS from SPMS, necessitating further investigation.

Serum biomarkers such as neurofilament light chain (sNfL) have been widely studied in multiple sclerosis and other neurodegenerative conditions. In particular, the sNfL has been shown to correlate with neurological disease severity and axonal injury across various disorders, including Wilson’s disease [12,13]. These findings suggest that serum neurofilaments may represent a cross-disease marker of neurodegeneration, supporting their inclusion in studies exploring MS progression.

Therefore, this study aims to investigate whether the values of the studied biomarkers differ between active relapsing–remitting multiple sclerosis (RRMS) patients and secondary progressive multiple sclerosis (SPMS) patients. Additionally, it examines whether the studied biomarkers have the same prognostic value, particularly the correlation between SEMA3A and neurofilaments, in relation to the Expanded Disability Status Scale (EDSS) in MS patients. This study further explores whether the relationship between biomarker levels and EDSS differs between RRMS and SPMS patients and whether the interrelationships among the biomarkers vary between these two patient groups.

## 2. Results

### 2.1. RRMS and SPMS Patient Group Characteristics: Description and Comparison

The RR patient group (*n* = 63) was younger at the time of measurement, with a mean age of 44.71 years (standard deviation = 12.23), compared to the SPMS group (*n* = 55), which had a mean age of 60.85 years (7.59). The proportion of female patients is 71.4% in the RR group and 63.6% in the SPMS group.

The two groups showed significant differences in clinical characteristics. The EDSS scores in the RR group averaged 3.25 (1.46), whereas in the SPMS group, they averaged 6.64 (0.61). These scores correspond to the mean disease duration of 7.75 years (8.12) in the RR group and 26.27 years (9.68) in the SPMS group. On average, SPMS patients converted from RRMS to SPMS after approximately 13 years (calculated as mean MS duration minus mean SPMS duration). Due to the clinical differences between the groups, including disease duration and severity, their current and previous treatments also differed. At the time of measurement, the most common treatments in the RR group were natalizumab (22.2%) and ofatumumab (22.2%), followed by INF-ß (15.9%), with 49.2% of RR patients receiving first-line treatment. In contrast, only 5.5% of SPMS patients were on first-line therapy, with the most frequent treatments being 1 g intravenous methylprednisolone (IVMP) per month (34.5%), 15 g intravenous immunoglobulin (IVIG) every month (16.4%), and siponimod (10.9%).

There were no statistically significant (at the 5% significance level) differences between the groups in any of the studied biomarkers. The given values are in the following format: (mean (SD); median). The levels of progranulin were 6.16 (5.67); 4.20 in RR patients and 4.93 (4.26); 3.66 in SPMS patients. IL-6 levels were 392.40 (525.75); 108.30 in the RR group and 453.78 (547.09); 136.10 in the SPMS group. SEMA3A levels were 13.60 (10.08); 11.93 in the RR group and 16.46 (15.28); 11.47 in the SPMS group. Neurofilament values were centered around 47.00 (27.15); 44.15 in the RR group and 56.15 (47.29); 44.94 in the SPMS group. A comprehensive summary of the observed characteristics and statistical comparisons (*p*-values) is available in Table 1 for numerical variables. The validation analysis confirms that there are no statistically significant differences between SPMS and RR MS groups in studied biomarkers after adjustment for confounding factors such as age and MS duration; please see Appendix A for more details.

### 2.2. Mutual Correlation Between Biomarkers and Their Association with Clinical Disability (EDSS)

The results of the correlation analysis indicate a linear relationship between SEMA3A and neurofilaments (Pearson’s correlation coefficient = 0.36, *p* value = 0.007) and between progranulin and IL-6 (0.35, *p*-value = 0.008) in the SPMS cohort. In the RR population, neither these associations nor any other correlations among the studied biomarkers were statistically significant. The visualization of the relationship between SEMA3A and neurofilaments for both groups is presented in Figure 1, while the relationship between progranulin and IL-6 is illustrated in Figure 2.

The validation analysis adjusting the correlation for the influential factors, such as age and MS duration, of MS confirms that the strongest relationship is between SEMA3A and neurofilaments (multiple correlation coefficient = 0.44) and between progranulin and IL-6 (0.39) in the SPMS cohort and the strongest relationship is between SEMA3A and neurofilaments (0.25) in RR MS patients. All multiple correlation coefficients are shown in Appendix A.

No statistically significant association between EDSS and any of the studied biomarkers was observed in either the SPMS or RR groups. All correlation coefficients are presented in Table 2 for the RR group and Table 3 for the SPMS group. Adjusted *p*-values are provided in parentheses above the diagonal. Validation analysis adjusting the correlation for the influential factors, such as age and duration, of MS shows that the strongest mutual relationship is between EDSS and neurofilaments (0.21) in RR MS patients, and in SPMS patients, EDSS is most strongly correlated with SEMA3A (0.32) and with progranulin (0.21); see Appendix A.

## 3. Discussion

### 3.1. Demographic and Clinical Differences Between RRMS and SPMS

Our analysis confirmed significant differences between patients with relapsing–remitting (RRMS) and secondary progressive (SPMS) multiple sclerosis in terms of demographic and clinical characteristics. As expected, SPMS patients were significantly older than RRMS patients (60.85 vs. 44.71 years, *p* < 0.001), reflecting the natural progression of the disease. Disease duration was also significantly longer in the SPMS group (26.27 vs. 7.75 years, *p* < 0.001), which corresponded to a higher degree of disability measured by the Expanded Disability Status Scale (EDSS) (6.64 vs. 3.25; *p* < 0.001). These differences underscore the transition from the relapsing phase to the chronic progressive stage of MS.

### 3.2. Comparison of Absolute Biomarker Levels Between Groups

In contrast to demographic and clinical parameters, the absolute levels of all investigated biomarkers—progranulin, IL-6, SEMA3A, and neurofilaments—did not differ significantly between the RRMS and SPMS groups (all *p* > 0.05). Although higher levels of neurofilaments were anticipated in SPMS patients due to advanced neurodegeneration, the observed values were not significantly different (56.15 vs. 47.00; *p* = 0.658). Similarly, IL-6 levels did not significantly differ between SPMS and RRMS patients (453.78 vs. 392.40; *p* = 0.738).

These negative findings persisted even after adjusting for age and disease duration. This suggests that, in a cross-sectional setting, these biomarkers may have limited discriminative value between MS phenotypes. Alternatively, their expression may be influenced by other factors such as pharmacotherapy, inter-individual variability, or genetic predispositions.

### 3.3. Intra-Group Correlations of Biomarkers in SPMS and RRMS

Despite the lack of between-group differences, relevant intra-group associations emerged. In SPMS patients, a significant positive correlation was observed between SEMA3A and neurofilament levels (r = 0.359; *p* = 0.007), whereas no such correlation was present in RRMS (r = 0.191; *p* = 0.137). SEMA3A, an axonal guidance molecule that impairs regeneration, has been implicated in the pathogenesis of neurodegeneration in multiple neurological diseases. Its observed correlation with neurofilaments in SPMS suggests a mechanistic link to ongoing neuroaxonal injury in progressive MS.

Similarly, a significant correlation was identified between progranulin and IL-6 in SPMS patients (r = 0.354; *p* = 0.008), not observed in RRMS (r = 0.073; *p* = 0.573). Progranulin plays a dual role in neuroprotection and immunomodulation, while IL-6 is a key pro-inflammatory cytokine. Their co-expression in SPMS may reflect a distinctive immuno-neurodegenerative profile characteristic of the progressive disease stage.

### 3.4. Relationship Between Biomarkers and Disability (EDSS)

None of the studied biomarkers showed a statistically significant correlation with EDSS in either SPMS or RRMS. The strongest observed correlations were for SEMA3A (r = −0.22; *p* = 0.109) and progranulin (r = −0.02; *p* = 0.91) in SPMS and neurofilaments (r = 0.20; *p* = 0.12) in RRMS. However, none of these associations reached statistical significance.

Multivariate correlation analysis, adjusted for age and disease duration, yielded slightly higher coefficients (e.g., SEMA3A–EDSS in SPMS = 0.32), but these too remained statistically non-significant. This suggests that the studied biomarkers, while mechanistically plausible, may not robustly reflect functional disability in a cross-sectional setting.

These findings contrast with previous research, where the sNfL has been consistently associated with clinical disability and disease severity in both MS and other neurological disorders. For example, studies in Wilson’s disease have demonstrated that elevated sNfL levels reflect neuroaxonal damage and correlate with functional decline [12,13]. The lack of such correlations in our cohort may reflect a combination of limited statistical power, subgroup heterogeneity, and the limitations of cross-sectional designs in capturing dynamic disease trajectories. Therefore, our findings do not support the use of these serum biomarkers as stand-alone indicators of disability severity in MS at this stage.

### 3.5. Limitations and Future Directions

A key limitation of our study is its relatively small sample size, which may have affected the statistical power of our findings. The lack of significant correlations between biomarkers and EDSS may also reflect a temporal mismatch: for instance, neurofilaments may be more indicative of acute axonal damage than cumulative disability.

The post hoc power analysis of the key correlations revealed that only the strongest associations observed in the SPMS cohort—namely between SEMA3A and neurofilaments, and between progranulin and IL-6—had sufficient statistical power (≥0.85), whereas correlations in the RRMS cohort were underpowered (≤0.45). This highlights the possibility that additional biologically meaningful associations may have been missed due to insufficient sample size, especially in subgroup analyses.

Another major limitation is the absence of a healthy control cohort, which prevents interpretation of absolute biomarker levels in the context of physiological norms. Although intra-MS comparisons are informative, it remains unclear whether the observed biomarker concentrations are elevated or suppressed relative to healthy individuals.

Furthermore, the cross-sectional design of this study does not allow conclusions to be drawn about temporal or causal relationships. Thus, while certain biomarker associations are intriguing, their prognostic value cannot be confirmed in this study design. Future research should incorporate longitudinal follow-up, serial biomarker assessments, and integration of imaging techniques such as MRI, ideally with age- and sex-matched healthy controls, to provide a more comprehensive understanding of MS pathophysiology and progression.

In addition, future studies should consider integrating serum biomarkers with neuroradiological methods such as MRI-based volumetry, diffusion tensor imaging, or cortical thickness analysis. A multimodal approach may improve sensitivity to disease-related changes and allow for the better stratification of MS subtypes and their clinical trajectories.

## 4. Materials and Methods

### 4.1. Study Population

This observational study was conducted from May 2023 to November 2024. In total, 118 participants were enrolled. The study group comprised 55 patients with secondary progressive multiple sclerosis (SPMS) and 63 patients with relapsing–remitting multiple sclerosis (RRMS).

For patients with SPMS, the inclusion criteria were as follows: Participants had to be between 18 and 75 years of age at the time of signing the informed consent. They had to have been previously diagnosed with RRMS in accordance with the 2017 revised McDonald criteria [14] and currently diagnosed with SPMS based on the clinical course criteria [15] (Lublin, 1996), revised in 2013 [16]. Additionally, they had to have been free of clinical relapses for at least 24 months.

Participants were required to discontinue treatment with any of the following medications prior to study entry: intravenous immunoglobulin, dimethyl fumarate, ponesimod, siponimod, ozanimod, fingolimod, teriflunomide, azathioprine, mycophenolate mofetil, methotrexate, and B-cell-depleting therapies (such as ocrelizumab and rituximab) at least 12 months before enrollment. Treatment with mitoxantrone, cyclophosphamide, cladribine, cyclosporine, and alemtuzumab had to be suspended at least 2 years prior to study entry. Additionally, treatment with methylprednisolone, glatiramer acetate, and interferon beta had to be discontinued at least 3 months before participation.

Patients with RRMS were treated with standard disease-modifying therapy (DMT) in accordance with the reimbursement criteria valid in the Czech Republic. The inclusion criteria for RRMS patients required participants to be between 18 and 75 years of age at the time of signing the informed consent and to be diagnosed with RRMS in accordance with the 2017 revised McDonald criteria [14]. Furthermore, they had to have experienced at least one clinical relapse in the three months preceding laboratory examination and had to have been treated with DMT for no more than three years.

All patients provided written informed consent. The study protocol was approved by the Ethics Committee of the University Hospital Hradec Králové (reference number: 201801S08P).

### 4.2. Primary and Secondary Objectives and Endpoints

The primary objective of this study is to describe and statistically compare the differences in the levels of the studied biomarkers between RRMS and SPMS patients. The primary endpoint focuses on reporting and statistically analyzing the measured levels of progranulin, interleukin-6 (IL-6), semaphorin 3A (SEMA3A), and neurofilaments while assessing the statistical differences in biomarker levels between RRMS and SPMS patients.

The secondary objectives include describing and statistically comparing demographic and clinical characteristics between RRMS and SPMS patients. Furthermore, this study seeks to investigate potential interrelationships among the studied biomarkers and their correlation with EDSS, analyzed separately for RRMS and SPMS patients. Regarding secondary endpoints, this study aims to describe and statistically compare demographic and clinical characteristics between RRMS and SPMS patients, while also assessing the degree of linear relationships between these parameters using correlation analysis. The investigated characteristics include current age, age at diagnosis, gender, EDSS score, MS duration (in years), SPMS duration (in years, if applicable), and MS-related therapy, both current and past.

### 4.3. Determination of Serum Analytes

The serum concentrations of the investigated analytes were measured using a sandwich enzyme-linked immunosorbent assay (ELISA) technique. The employed ELISA kits were as follows: Human IL-6 (RayBiotech, Peachtree Corners, GA, USA), Human Phosphorylated Neurofilament (BioVendor, Brno, Czech Republic), Human Progranulin (RayBiotech, Peachtree Corners, GA, USA), and Human Semaphorin 3A (Cloud-Clone Corp, Katy, TX, USA). All assays were performed according to the manufacturer’s instructions.

The minimum detectable dose for the Human IL-6 ELISA kit was 3 pg/mL, with data expressed in pg/mL. For the Human Phosphorylated Neurofilament ELISA kit, the minimum detectable dose was 23.5 pg/mL, and data were also expressed in pg/mL. The Human Progranulin ELISA kit had a minimum detectable dose of 0.1 ng/mL, with data reported in ng/mL, while the Human Semaphorin 3A ELISA kit had a minimum detectable dose of 0.63 ng/mL, with data expressed in ng/mL.

Absorbance values were measured at 450 nm using a Multiskan™ FC Microplate Photometer (Thermo Fisher Scientific, Waltham, MA, USA) equipped with SkanIt™ Software for Microplate Readers (version 5.0, Thermo Fisher Scientific) (Thermo Fisher Scientific, Waltham, MA, USA). Serum samples for IL-6, progranulin, and semaphorin 3A assays were analyzed undiluted, whereas samples for neurofilament determination were diluted threefold based on preliminary test results.

### 4.4. Statistical Analysis

The studied biomarkers and clinical and demographic characteristics of patients are presented for the overall RRMS cohort (n = 118) as well as separately for the RRMS (RR = 63) and SPMS (SPMS = 55) subgroups. Continuous variables are reported as mean (standard deviation) and median [lower quartile, upper quartile]. Due to the skewed distribution of most variables, assessed graphically and using the Lilliefors normality test, between-group differences were tested using non-parametric two-sample *t*-tests. Categorical variables are reported as frequency (%), with percentages calculated based on the total number of non-missing values overall and within each group. Differences between RR and SPMS for categorical variables were tested using the Chi-square test of independence with continuity correction.

Correlation analysis was conducted to evaluate potential mutual (linear) relationships between the studied parameters using Pearson’s pairwise correlation coefficient (r). The statistical significance of correlations (null hypothesis: r = 0) was assessed with the Benjamini–Hochberg correction for multiple testing.

Multivariate regression analysis was performed to validate the existence of studied relationships after adjustment for confounding factors: age and duration of disease. The following regression model was fitted to validate differences between groups in studied biomarkers: *Y* = *group (reference group = RR MS)* + *age* + *disease duration*, using lm () function. To assess the correlations between biomarkers and EDSS after adjustment for confounding factors (age, MS duration) separately for SPMS and RR MS groups, the multiple correlation coefficients were calculated as follows. A multiple regression model *Y* = *X* + *age* + *disease duration* was fitted. Having created the above model, the observed values with the predicted values for the same variable were correlated.

All statistical tests were performed using a two-tailed alternative hypothesis with a significance level (type I error) set at 5%. All statistical analyses were conducted using the R statistical software (https://www.r-project.org, version 4.3.1, released on 16 June 2023).

## 5. Conclusions

Our findings highlight significant clinical and demographic differences between RR and SPMS MS patients, reflecting the natural progression of the disease. While no significant differences were observed in individual biomarker levels between the two groups, the identified correlations suggest that specific biomarkers, such as SEMA3A and neurofilaments, may be associated with neurodegeneration in the progressive phase of MS. Similarly, the strong correlation between progranulin and IL-6 in SPMS patients underscores the potential interplay between chronic inflammation and neurodegenerative processes. These results suggest that, while single biomarker assessments may have limited utility in distinguishing MS phenotypes, a combination of inflammatory and neurodegenerative markers may provide valuable insights into disease progression.

## Figures and Tables

**Figure 1 ijms-26-08750-f001:**
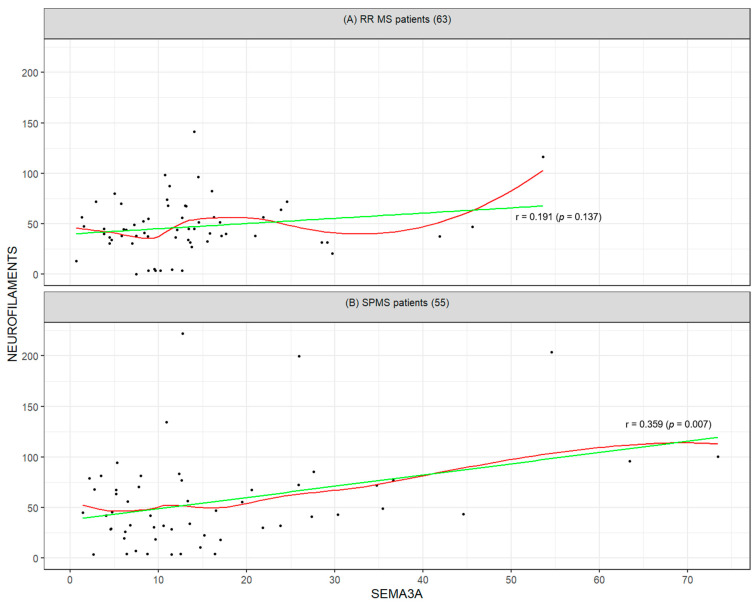
Visualization of the relationship between SEMA3A and neurofilaments for (**A**) RRMS patients (*n* = 63) and (**B**) SPMS patients (*n* = 55). The green line shows the estimated linear correlation (linear regression model) and the red curve shows the non-parametric (LOESS) correlation estimate.

**Figure 2 ijms-26-08750-f002:**
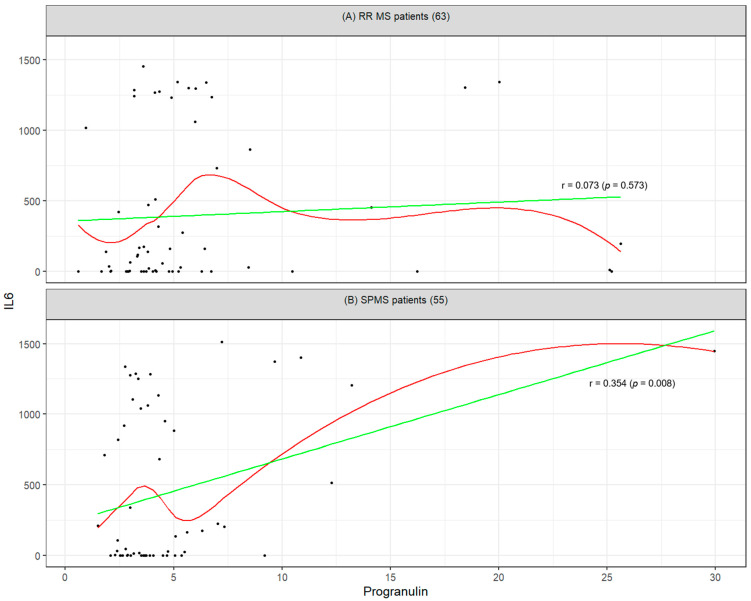
Visualization of the relationship between progranulin and IL-6 for (**A**) RRMS patients (*n* = 63) and (**B**) SPMS patients (*n* = 55). The green line shows the estimated linear correlation (linear regression model) and the red curve shows the non-parametric (LOESS) correlation estimate.

**Table 1 ijms-26-08750-t001:** Descriptive statistics by MS type for investigated numerical variables. The continuous variables are reported as mean (standard deviation—SD), median [lower quartile—p25; upper quartile—p75]. Between-group differences were tested by non-parametric two-sample *t*-tests with two-tailed alternative hypotheses.

Variable	Overall (118)		SPMS (55)		RR (63)		SPMS vs. RR
	Mean (SD)	Med [p25, p75]	Mean (SD)	Med [p25, p75]	Mean (SD)	Med [p25, p75]	*p*-Value
Progranulin (ng/mL)	5.59 (5.08)	3.98 [2.99, 5.67]	4.93 (4.26)	3.66 [2.85, 5.07]	6.16 (5.67)	4.20 [3.24, 6.36]	0.136
IL-6 (pg/mL)	421.01 (534.38)	114.05 [0, 944.65]	453.78 (547.09)	136.10 [0.00, 996.25]	392.40 (525.75)	108.30 [0.00, 800.05]	0.738
SEMA3A (ng/mL)	14.93 (12.79)	11.73 [6.57, 17.02]	16.46 (15.28)	11.47 [6.29, 21.20]	13.60 (10.08)	11.93 [7.35, 15.94]	0.802
Neurofilaments (pg/mL)	51.26 (38)	44.25 [30.88, 68.05]	56.15 (47.29)	44.94 [28.63, 74.75]	47.00 (27.15)	44.15 [33.11, 56.69]	0.658
Current age	53.02 (13.13)	53.29 [44.58, 63.48]	61.67 (7.60)	63.04 [56.08, 67.56]	45.47 (12.28)	45.89 [34.94, 51.69]	<0.001
Age	52.24 (13.08)	52.5 [44.0, 63.0]	60.85 (7.59)	62.00 [55.50, 66.50]	44.71 (12.23)	45.00 [34.00, 51.00]	<0.001
EDSS	4.83 (2.04)	5.25 [3.0, 6.5]	6.64 (0.61)	6.50 [6.50, 7.00]	3.25 (1.46)	3.00 [2.00, 4.50]	<0.001
MS duration (in years)	16.38 (12.82)	14.0 [4.0, 27.0]	26.27 (9.68)	26.00 [21.00, 32.50]	7.75 (8.12)	5.00 [2.00, 9.50]	<0.001
CH/P duration (in years)	6.2 (8.12)	0.0 [0.0, 12.0]	13.18 (6.92)	13.00 [9.50, 17.00]	0.00 (0.00)	0.00 [0.00, 0.00]	<0.001

**Table 2 ijms-26-08750-t002:** Pairwise correlations between studied outcomes and EDSS for the RRMS group. The correlation matrix displays Pearson’s pairwise correlation coefficients. *p*-values from statistical testing (two-sided) of the significance of the correlation coefficient are assigned to the correlation coefficient in parentheses. Correlation coefficients adjusted for multiple testing are reported (in brackets) above the diagonal.

	Progranulin	IL6	SEMA3A	NEUROFILAMENTS	EDSS
**Progranulin**	1 (0)	0.07 (0.825)	−0.13 (0.635)	−0.06 (0.825)	−0.06 (0.825)
**IL6**	0.07 (0.573)	1 (0)	0.01 (0.945)	−0.13 (0.635)	−0.13 (0.635)
**SEMA3A**	−0.13 (0.302)	0.01 (0.945)	1 (0)	0.19 (0.635)	−0.01 (0.945)
**NEUROFILAMENTS**	−0.06 (0.625)	−0.13 (0.308)	0.19 (0.137)	1 (0)	0.2 (0.635)
**EDSS**	−0.06 (0.66)	−0.13 (0.318)	−0.01 (0.916)	0.2 (0.12)	1 (0)

**Table 3 ijms-26-08750-t003:** Pairwise correlations between studied outcomes and EDSS for the SPMS group. The correlation matrix displays Pearson’s pairwise correlation coefficients. *p*-values from statistical testing (two-sided) of the significance of the correlation coefficient are assigned to the correlation coefficient in parentheses. Correlation coefficients adjusted for multiple testing are reported (in brackets) above the diagonal.

	Progranulin	IL6	SEMA3A	NEUROFILAMENTS	EDSS
**Progranulin**	1 (0)	0.35 (0.04)	0.17 (0.417)	0.24 (0.261)	−0.02 (0.91)
**IL6**	0.35 (0.008)	1 (0)	0.16 (0.417)	−0.04 (0.827)	−0.08 (0.723)
**SEMA3A**	0.17 (0.225)	0.16 (0.25)	1 (0)	0.36 (0.04)	−0.22 (0.272)
**NEUROFILAMENTS**	0.24 (0.078)	−0.04 (0.745)	0.36 (0.007)	1 (0)	−0.11 (0.625)
**EDSS**	−0.02 (0.91)	−0.08 (0.578)	−0.22 (0.109)	−0.11 (0.437)	1 (0)

## Data Availability

The original contributions presented in this study are included in the article. Further inquiries can be directed to the corresponding author.

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
