# Peer review of "Exploring the Prognostic Role of Neurofilaments and SEMA3A in Multiple Sclerosis Progression"

_ijms, 2025, doi:10.3390/ijms26178750_

Round 1

Reviewer 1 Report

Comments and Suggestions for Authors

The manuscript addresses an important clinical question regarding biomarkers for distinguishing between RRMS and SPMS. The study is well-written, methods are generally sound, and the statistical approach is described in detail. However, several methodological and interpretational issues need to be addressed before the work can be considered for publication.

Major comments:

  1. Sample size & power – The cohort (63 RRMS, 55 SPMS) may be underpowered to detect moderate differences. Please provide a priori sample size justification or post-hoc power analysis.

  2. Confounding variables – Age, disease duration, and treatment regimens differ substantially between groups. Adjusting for these via multivariable analysis would improve robustness.

  3. Control group – The absence of a healthy control cohort limits interpretation of absolute biomarker levels; this should be acknowledged as a major limitation.

  4. Therapy effects – Current and previous treatments vary significantly between groups and may influence biomarker levels. Please discuss or adjust for this factor.

  5. Interpretation of negative findings – Statements implying prognostic potential should be clearly separated from statistically supported results.

  6. Cross-sectional design – This design does not allow conclusions about prognostic value; highlight this limitation and suggest future longitudinal validation.

    Minor comments:

    1. Correct repeated “2.1” subsection numbering in Materials and Methods.

    2. Define IVMP and IVIG at first mention.

    3. Fix minor table formatting inconsistencies (spacing, p-value presentation).

    4. Add correlation coefficients and p-values directly into Figures 1 and 2.

    5. Consider restructuring Discussion to separate significant correlations from non-significant group comparisons for clarity.

Author Response

Dear Reviewer,

We are writing to you about our article called Exploring the Prognostic Role of Neurofilaments and SEMA3A in Multiple Sclerosis Progression, which needed revisions in order to be considered for the publication.

We thank you for the constructive and justified comments. We tried hard to incorporate all the comments (consult below) in a very thorough manner. Overall, thanks to the review, the article has been improved.

Major comments:

  1. Sample size & power – The cohort (63 RRMS, 55 SPMS) may be underpowered to detect moderate differences. Please provide a priori sample size justification or post-hoc power analysis.

Response: A priori power analysis was not performed for this observational exploratory study. Post-hoc power analysis (with alpha = 0.05) for selected correlations - correlation coefficients r=0.191, n=63 (Figures 1A); 0.359, 55 (Fig. 1B); 0.073, 63 (Fig. 2A); 0.354, 55 (Fig. 2B) determined the following statistical power: 0.45; 0.86; 0.14; 0.85. The estimates of the correlation coefficients between SEMA3A and Neurofilaments and between Progranulin and IL6 for the SPMS cohort have sufficient statistical power.
We suggest adding these post hoc analysis results to the Discussion (Section 3.3)

  1. Confounding variables – Age, disease duration, and treatment regimens differ substantially between groups. Adjusting for these via multivariable analysis would improve robustness.

Response: We thank you for the recommendation to validate the results of the analysis and adjust for influential factors. We performed a multivariate regression analysis to verify the significance of the differences between groups (Table 1) and calculated multiple correlation coefficients to validate the significance of the correlation coefficients. We have added to the manuscript the interpretation of the results of this validation in sections 2.1 and 2.2, extended discussion in section 3.3, a summary of the results as Supplementary Material 1 and 2, and a description of the methodology in Section 4.4. We added age and duration of disease as confounding factors to the regression models and correlation analysis. We did not consider treatment history due to the large inter-patients variability of treatment patterns and the small number of patients. However, the duration of the disease reflects to some extent differences in treatment patterns.

  1. Control group – The absence of a healthy control cohort limits interpretation of absolute biomarker levels; this should be acknowledged as a major limitation.

We agree with the reviewer that the absence of a healthy control group is a relevant limitation that reduces the interpretability of absolute biomarker values. Our primary objective was to compare RRMS and SPMS cohorts to identify relative differences and potential correlations between biomarkers and clinical parameters within the disease spectrum. However, without a healthy control group, it is not possible to determine whether the measured levels are elevated, reduced, or within normal limits in either MS subgroup. Accordingly, we have updated Limitations paragraph in the Discussion section.

  1. Therapy effects – Current and previous treatments vary significantly between groups and may influence biomarker levels. Please discuss or adjust for this factor.

Response: The effect of treatment sequence (treatment history and current treatment) was not directly included in the statistical calculations. This is due to sparse data due to high inter-patient variability of sequences and small number of patients. However, MS duration, which was included as an confounding factor in the validation analysis, carries partly similar information.

  1. Interpretation of negative findings – Statements implying prognostic potential should be clearly separated from statistically supported results.

We thank the reviewer for this important clarification. We fully agree that interpretations implying prognostic potential should be clearly separated from statistically supported findings. Accordingly, we have revised the Discussion section to strictly distinguish between group comparisons, intra-group correlations, and exploratory trends. Furthermore, we have removed or rephrased speculative statements and explicitly acknowledged that the cross-sectional design precludes prognostic conclusions. We believe that the revised structure and language now reflect an appropriately cautious interpretation of our data.

  1. Cross-sectional design – This design does not allow conclusions about prognostic value; highlight this limitation and suggest future longitudinal validation.

We thank the reviewer for this important observation. We fully agree that the cross-sectional design of our study precludes any definitive conclusions regarding the prognostic value of the investigated biomarkers. While our findings highlight associations between selected biomarkers and disease status within a defined timepoint, they do not establish temporal or causal relationships. We have revised the Discussion sections to acknowledge this limitation explicitly and emphasize the need for longitudinal validation in future studies.

Minor comments:

    1. Correct repeated “2.1” subsection numbering in Materials and Methods.

Response: Corrected.

    1. Define IVMP and IVIG at first mention.

We thank the reviewer for this observation. We have revised the manuscript to ensure that the abbreviations IVMP (intravenous methylprednisolone) and IVIG (intravenous immunoglobulin) are defined upon first mention in the text. These definitions have now been included in the Results section, specifically in the paragraph describing treatment regimens in the RRMS and SPMS cohorts.

Fix minor table formatting inconsistencies (spacing, p-value presentation). corrected

    1. Add correlation coefficients and p-values directly into Figures 1 and 2.

Response: In Figures 1A, 1B, 2A, 2B, the correlation coefficients and their corresponding p-values ​​are mentioned (in the header).

    1. Consider restructuring Discussion to separate significant correlations from non-significant group comparisons for clarity.

We thank the reviewer for this helpful suggestion. We agree that separating the discussion of group comparisons (RRMS vs. SPMS) from the analysis of significant correlations improves the clarity and logical flow of the manuscript. Accordingly, we have restructured the Discussion section into distinct subsections: one focusing on group-level differences (or lack thereof), and another addressing significant intra-group biomarker correlations and their potential mechanistic implications. We believe this improves the interpretability and readability of our findings.

Reviewer 2 Report

Comments and Suggestions for Authors

The study aimned to find the biomarkers which would be helpfull in differential diagnosis of SM RR and SP MS. Tha authors used progranulin, interleukin-6 (IL-6), semaphorin 3A (SEMA3A), and neurofilaments analysis and didn't find any association, what could be suspeted as SPMS occure often in further studies of MS, however severity of disease (neurological) should correlate with sNfL .

Th interrestingly biomarkers didn't correlate with severity of disease (EDSS) only each other. (low number of patients). Based on studies of sNfL in several disorders it usually correlate with severity of neurological disease (see e.g studies with sNFl and Wilson's disease - Ziemssen, et al 2022 and 2023) , the authors should disscuss more extensively thier results (especially lack of correlations with EDSS) always in introduction few more sentences according to significance of this biomarkers in other neurological disorders should be discusssed. The exact results no asscociation between serum biomarkers and form of MS is interesting and worth further studies with other serum biomerkers as well as neuroradiological (the authors should add future directions of such stdueis) 

Author Response

Dear Reviewer,

We are writing to you about our article called Exploring the Prognostic Role of Neurofilaments and SEMA3A in Multiple Sclerosis Progression, which needed revisions in order to be considered for the publication.

We thank you for the constructive and justified comments. We tried hard to incorporate all the comments (consult below) in a very thorough manner. Overall, thanks to the review, the article has been improved.

-------------------------------------------------------------------------------------------------------------------------------------

The study aimned to find the biomarkers which would be helpfull in differential diagnosis of SM RR and SP MS. Tha authors used progranulin, interleukin-6 (IL-6), semaphorin 3A (SEMA3A), and neurofilaments analysis and didn't find any association, what could be suspeted as SPMS occure often in further studies of MS, however severity of disease (neurological) should correlate with sNfL .

Th interrestingly biomarkers didn't correlate with severity of disease (EDSS) only each other. (low number of patients). Based on studies of sNfL in several disorders it usually correlate with severity of neurological disease (see e.g studies with sNFl and Wilson's disease - Ziemssen, et al 2022 and 2023) , the authors should disscuss more extensively thier results (especially lack of correlations with EDSS) always in introduction few more sentences according to significance of this biomarkers in other neurological disorders should be discusssed. The exact results no asscociation between serum biomarkers and form of MS is interesting and worth further studies with other serum biomerkers as well as neuroradiological (the authors should add future directions of such stdueis)

We thank the reviewer for this thoughtful and comprehensive comment. We agree that serum neurofilament light chain (sNfL) has been demonstrated in various neurological diseases—including multiple sclerosis and Wilson’s disease (as shown in Ziemssen et al., 2022 and 2023)—to correlate with clinical severity and neuroaxonal damage. In our study, the absence of a statistically significant correlation between sNfL and EDSS is indeed unexpected. As suggested, this may be attributed to the relatively small sample size, particularly within subgroups, and to the cross-sectional nature of our design, which limits the temporal resolution of disability progression. We have revised the Discussion to more extensively address this discrepancy and to place our findings in the context of previous literature on sNfL and disease severity. Additionally, we have added a short paragraph in the Introduction summarizing the established relevance of these biomarkers in other neurological disorders Furthermore, we acknowledge that our negative findings do not invalidate the potential utility of serum biomarkers in MS monitoring. On the contrary, they underscore the need for larger, longitudinal, multimodal studies combining serum markers with neuroradiological metrics (e.g., volumetric MRI or diffusion imaging). We have added this perspective to the “Future Directions” subsection.

Round 2

Reviewer 2 Report

Comments and Suggestions for Authors

The authors corrected article. I recommend to accept the paper. I have no more comments